# Detection of Hepatitis E Virus Genotype 3 in Feces of Capybaras (Hydrochoeris hydrochaeris) in Brazil

**DOI:** 10.3390/v15020335

**Published:** 2023-01-24

**Authors:** Lia Cunha, Adriana Luchs, Lais S. Azevedo, Vanessa C. M. Silva, Marcilio F. Lemos, Antonio C. Costa, Adriana P. Compri, Yasmin França, Ellen Viana, Fernanda Malta, Roberta S. Medeiros, Raquel Guiducci, Simone G. Morillo, Michele S. Gomes-Gouvea, Deyvid Amgarten, João R. R. Pinho, Regina C. Moreira

**Affiliations:** 1Instituto Adolfo Lutz, Sao Paulo 01246-902, Brazil; 2Instituto de Medicina Tropical de São Paulo, Universidade de São Paulo, São Paulo 05403-000, Brazil; 3Hospital Israelita Albert Einstein, Sao Paulo 05652-900, Brazil

**Keywords:** Hepatitis E virus, genotype 3, public health, phylogeny, zoonosis

## Abstract

Hepatitis E virus (HEV) is an emerging zoonotic pathogen associated with relevant public health issues. The aim of this study was to investigate HEV presence in free-living capybaras inhabiting urban parks in São Paulo state, Brazil. Molecular characterization of HEV positive samples was undertaken to elucidate the genetic diversity of the virus in these animals. A total of 337 fecal samples were screened for HEV using RT-qPCR and further confirmed by conventional nested RT-PCR. HEV genotype and subtype were determined using Sanger and next-generation sequencing. HEV was detected in one specimen (0.3%) and assigned as HEV-3f. The IAL-HEV_921 HEV-3f strain showed a close relationship to European swine, wild boar and human strains (90.7–93.2% nt), suggesting an interspecies transmission. Molecular epidemiology of HEV is poorly investigated in Brazil; subtype 3f has been reported in swine. This is the first report of HEV detected in capybara stool samples worldwide.

## 1. Introduction

Hepatitis E is an infectious disease caused by the hepatitis E virus (HEV), with an estimated 20 million cases each year [1]. HEV is the major cause of acute hepatitis globally and is associated with large outbreaks in regions of endemicity, mainly in poor resource areas including Asia, the Middle East and Africa [2,3,4]. Hepatitis E is usually a mild self-limiting disease; however, it may be fatal among pregnant women, in which mortality rates could reaches 10–25% during the third trimester, or become chronic in immunocompromised patients [5].

HEV is a small virus with a 7.2 kb positive-sense RNA genome. Particles are spherical with a diameter of 27–34 nm. HEV circulates as quasi-enveloped virions in blood and cell culture supernatants; however, HEV is secreted as a non-enveloped virus in feces and bile [6]. Its genomic RNA is polyadenylated and contains three ORFs which span the genome; ORF1 encodes a non-structural polyprotein with multiple functional domains, including those for methyltransferase, protease, helicase, and polymerase. The viral capsid protein (CP) is encoded by ORF2 near the 3′-end. ORF3, which partially overlaps with the other two ORFs, codes for an immunogenic protein of unknown function [7].

HEV belongs to the Hepeviridae family (Riboviria realm), subfamily Orthohepivirinae, which is divided into four genera: Paslahepevirus (*Orthohepevirus* A); Avihepevirus (*Orthohepevirus* B); Rocahepevirus (*Orthohepevirus* C); and Chirohepevirus (*Orthohepevirus* D) [8]. Members of the genus *Paslahepevirus* have been identified in mammals including humans, pigs, deer, rabbits, camels, cattle, sheep, goats, mongooses, and bottlenose dolphins [4,9]. Members of the genus *Avihepevirus* are restricted to birds [10]. The genus *Rocahepevirus* includes viruses from rodents, eulipotyphlids and mustelids [11,12,13]. Members of the genus *Chirohepevirus* are limited to bats [14].

Considering the genus *Paslahepvirus* (*Orthohepevirus* A), members of the species *Paslahepevirus balayani* have been assigned to eight different genotypes, HEV-1 to HEV-8 [8,15]. HEV-1 to -4 are commonly associated with HEV infections in humans. HEV-1 and -2 are strictly related to outbreaks affecting humans. HEV-3 and -4 infect a wider spectrum of animals, including humans and other mammalian species, with domestic and wild pigs serving as the most important reservoirs [15,16]. HEV-5 and -6 have only been detected in wild boar in Japan [17]. HEV-7 has been detected in dromedary camels with only one report of human infection [18,19]. HEV-8 has only been detected in Bactrian camels in China [20].

Genotypes HEV-3 and HEV-4 are well known zoonotic viruses and have been frequently detected in humans, pigs and deer worldwide. Human infections are presumed to result from the consumption of raw or undercooked swine meat or derived products [2,21,22]. The host range of HEV-3 and HEV-4 have been intensively investigated, including their detection in goats, rats, dolphins, rabbits, cattle and sheep [23,24,25,26,27].

Considering the genus *Rocahepevirus* (*Orthohepevirus* C), the species *Rocahepevirus ratti* is frequently reported infecting rodents [11,12,13], but more recently was also identified in humans [28,29,30]. The zoonotic potential of Rocahepevirus *ratti* genotype C1 (HEV-C1) has been reinforced, since this genotype has been increasingly associated with infection and symptomatic disease (hepatitis) in humans across the globe [28,29].

HEV species *C* genotype 1 (HEV-C1) circulates in rats and is highly divergent from HEV-A. Previously, HEV-C1 was considered unable to infect humans, but the first case of human HEV-C1 infection was detected in Hong Kong. Sridhar (2021) et al. also observed that HEV-C could be persistent when infecting immunosuppressed patients [31]. Moreover, Rivera-Juarez (2022) described for the first time *Rocahepevirus* C in Europe evaluating cohorts of patients in Spain from 2018 to 2021, suggesting that *Orthohepevirus* C could be a new emergence in public health [32].

Rodents are the most species-rich order of mammals and well recognized as reservoirs associated with new virus’ emergence [33]. Recently, HEV (genera *Paslahepevirus* and *Rocahepevirus*) was reported infecting wild rats in the United States, Germany, Vietnam, China, Indonesia, United Kingdom and Italy [11,34,35,36,37,38,39].

Capybara (*Hydrochoerus hydrochaeris*) is the world’s largest rodent species, displaying a semi-aquatic habit and distributed throughout South America. Their high reproductive capacity, generalist feeding habits and minimal habitat quality requirements have contributed to overpopulation in numerous regions of Brazil. In addition, capybaras are regularly found in the vicinity of human populations in urban parks across the country [40,41]. *R*. *rickettsii* is transmitted to humans mainly by the tick *Amblyomma sculptum*, which uses the capybara as its main host [42]. Capybaras are also known to carry other potentially zoonotic agents, including *Leishmania* spp., *Leptospira* spp., *Trypanosoma* spp., *Salmonella* spp., *Toxoplasma gondii* and rabies virus [43,44].

In Brazil, HEV species *Paslahepevirus balayani* is known to be disseminated among swine populations as well as in derived pork products [45,46,47]. Nevertheless, the role of other mammals, including rodents, as potential hosts for HEV has been poorly investigated in the country [48,49]. HEV species *Paslahepevirus balayani* epidemiology is only partially understood in wild animals and monitoring the capybaras in areas cohabited by humans could help understand their importance as potential HEV reservoirs [50].

The aim of the present study was to investigate the HEV species *Paslahepevirus balayani’s* presence in free-living capybaras inhabiting urban parks in São Paulo state, Brazil. In addition, molecular characterization of positive samples was undertaken in order to gain insight into the genetic diversity of the virus in these animal populations.

## 2. Material and Methods

### 2.1. Sample Collection

A total of 337 fecal samples from free-living capybaras were collected between 2018 and 2020 from urban parks located in two municipalities of São Paulo state, Brazil. The (i) yacht club Santo Amaro (YCSA) is located on the shores of the Guarapiranga Reservoir; the (ii) Novo Rio Pinheiros bike lane is located on the banks of the Pinheiros River, both of them located in the southern region of São Paulo city; the (iii) Tietê Ecological Park (PET) is located in the eastern region of São Paulo city; and the (iv) municipal dam park is located in the eastern region of São José do Rio Preto city (Appendix A). The four parks inhabited by capybaras are characterized by low water-quality conditions and the pollutants are associated with anthropogenic environments, mostly emerging contaminants resulting from incomplete degradation in sewage treatment plants (https://cetesb.sp.gov.br/infoaguas/; accessed on 9 May 2022).

Fecal collection is a non-invasive procedure that does not require an animal’s capture. The capybara feces included here were obtained from natural evacuation. Fecal samples were collected fresh, preferably the portion that did not contact the soil or water and was not directly exposed to the sun. Each sample was considered tobelong to animals from the same batch (i.e., pool) because capybara are mammals that live in groups, making it impossible to identify the sample as belonging to a particular individual. Capybaras are very distinctive animals, and their feces are peculiar, standing on easily recognizable mounts. The stool samples obtained in natura were immediately stored at −20 °C. All frozen specimens were taken to the Enteric Disease Laboratory, Virology Center, Adolfo Lutz Institute, São Paulo, Brazil.

### 2.2. HEV Detection

Viral RNA was extracted from 10% of fecal suspensions using a QIAamp^®^ Viral RNA Mini Kit (QIAGEN, Valencia, CA, USA) according to the manufacturer’s instructions. HEV was screened by an in-house real-time RT-PCR targeting the overlapping region of ORF 2 and ORF 3 along with protocol, primers and probes previously defined by Jothikumar et al. [51], using 5 µL of each RNA in a reaction mixture with a final volume of 25 µL, using the GoTaq^®^ Probe 1-Step RT-qPCR System (Promega Corporation, Madison, WI, USA). Controls for RT-qPCR inhibitors or for RNA extraction efficiency are not available for capybara origin specimens. Therefore, viability of capybaras stools screened in the present study was checked by MALDI-TOF MS and spiked with experimental specimens with rotavirus and adenovirus reference strains (representing enteric RNA and DNA viruses, respectively) as described elsewhere [52]. The WHO international standard for HEV NAT assays (Paul- Ehrlich-Institut code 6329/10, Paul-Ehrlich-Institut, Langen, Germany) was included in order to evaluate the analytical specificity of the RT-qPCR assay as well as serving as a positive control reaction. Samples were considered positive when a Ct value ≤ 39 was obtained.

### 2.3. Amplification of Partial HEV ORF1 and ORF2

Positive-HEV samples were selected for molecular characterization. HEV was amplified directly from RNA extracted from the stool specimens and characterized by DNA sequencing and genetic analysis. The cDNA synthesis was performed using GoScript™ reverse transcriptase (Promega Corporation, WI, USA) according to the manufacturer’s recommendations. The regions encoding HEV ORF1 and ORF2 were used in semi-nested RT-PCRs following the primers and protocols described by Jhone et al. [11] and Wang et al. [53], respectively. The primers were designed to HEV ORF1 and ORF2 amplified fragments of ~450 bp and ~350 bp, respectively. Conventional semi-nested RT-PCR assays were monitored for false-positive or false-negative results using reaction controls containing a positive control, a negative control and sterile water.

### 2.4. HEV Typing

PCR amplicons were sequenced using a BigDye Terminator v3.1 Cycle Sequencing Kit (Applied Biosystems, Foster City, CA, USA) with the same primers used in the semi-nested PCRs targeting the ORF1 and ORF2 regions [11,53]. Dye-labeled products were sequenced in an ABI 3130 DNA Analyzer (Applied Biosystems). Sequences were edited with a Sequencher 4.7 (Gene Codes Corporation, Ann Harbor, MI). Hepatitis E Virus Genotyping Tool Version 0.1 (National Institute for Public Health and the Environment/RMVI, https://www.rivm.nl/mpf/typingtool/hev/; accessed on 14 September 2022) and hepatitis E virus genotyping tool (Genome Detective, https://www.genomedetective.com/app/typingtool/hev/; accessed on 14 September 2022) were used to assign the genotype of the study strains.

### 2.5. Nucleotide Sequencing of the Complete HEV Genomes

Samples that were successfully subtyped by the ORF1 and ORF2 typing nested RT-PCR and Sanger sequencing were subsequently subjected to next generation sequencing (NGS) to gain more information of the genome, ideally the full-genome sequence. RNA metagenomics sequencing (metatranscriptome) was performed to further characterize and recover whole HEV genomes. Briefly, total RNA was extracted from feces, DNA was removed by DNAse treatment and mammals rRNA was depleted. Reverse transcription was performed with a 2-step cDNA random synthesis, followed by PCR amplification, adapted from Greninger et al. [54]. DNA libraries were generated using Illumina Nextera XT (Illumina, CA) and then sequenced with 300-cycle sequencing kits in an Illumina NextSeq 550 instrument. Data generated as input were submitted to the Varsmetagen online platform (https://varsomics.com/varsmetagen/; accessed on 16 September 2022), where bioinformatics analyses and results interpretation were carried out. Varsmetagen virome pipeline performed the following steps: raw data quality control; host decontamination; first round of pathogen identification; short reads assembly; second round of identification with contigs; and finding confirmation through mapping and coverage metrics.

A total of 8.662 million raw short reads were generated by sequencing, with >97% of bases above a Q30 quality threshold. Low quality reads and reads mapping to humans were removed and remaining reads (27%) were used for downstream analyses. About 0.7% of reads were classified as viruses according to the Kraken2 tool (https://genomebiology.biomedcentral.com/articles/10.1186/s13059-019-1891-0; accessed on 16 September 2022) (Appendix A). Further validation was applied in order to rule out possible artifacts and viruses not related to the HEV, including plant and insect viruses present in samples due to diet or environmental contamination. The resulting singlets and contigs were analyzed using BLASTx to search for similarity to viral proteins in GenBank. The contigs were compared to the GenBank non-redundant nucleotide and protein databases (BLASTn and BLASTx). After identification of the viruses, reference template HEV sequences were used to map the full-length genomes with Geneious version 9.0.5 software (Biomatters Ltd. L2, Auckland, New Zealand).

### 2.6. Phylogenetic Analysis

In order to obtain more insightful information about the phylogenetic relationships of the HEV-3 genotype detected in this study, HEV-3 sequences obtained were aligned with a set of prototype sequences available in the GenBank database using the CLUSTAL W algorithm in the BioEdit sequence alignment editor software, version 7.0.5.2 (Ibis Therapeutics, Carlsbad, CA, USA). A maximum-likelihood tree was constructed for partial ORF1 and ORF2 genes. The best substitution models were selected based on the corrected Akaike information criterion (AICc) value as implemented in MEGA X [55]. The models used in this study were Tamura-Nei (TN93) +G +I for ORF1 and general time reversible (GTR) +G +I for ORF2. The statistical significance at the branch point was calculated with 1000 pseudo-replicate datasets. For the designation of subtypes, strains from GenBank were selected using lineages previously published by Smith et al. [15], Nicot et al. [56,57], and Sarchese et al. [58].

## 3. Results

Of the 337 stool samples tested, one specimen (0.3%) was positive for HEV by RT-qPCR (Ct values of 28.47 and 28.44 tested in duplicate). RNA representing the partial ORF1 and ORF2 regions could be successfully amplified by conventional nested RT-PCR, sequenced and genotyped from this sample, named here as IAL-HEV_921 strain. A 440 bp fragment of the ORF1 region and a 350 bp fragment of the ORF2 region generated a reliable classification as HEV genotype 3 according to data obtained by both HEV genotyping webtools used. The IAL-HEV_921 sample was collected in 2019 at the Tietê Ecological Park (PET).

NGS could recover two partial sequences of the ORF2 region, one 290 bp fragment (position 5777 to 6066 nt) and one 815 bp fragment (position 6267 to 7081 nt), totaling 1105 bp with 92% identical to isolate HEPAC-49 (MF444062), a HEV genotype 3 strain detected in a blood sample from a HEV-infected patient in France in 2009 [56] (Appendix A). About 1300 bp of the ORF2 region could be obtained from the IAL-HEV_921 strain by combining Sanger sequencing and NGS, and genotyped as HEV-3f according to the hepatitis E virus genotyping webtool Genome Detective.

To investigate the genetic relationships among contemporaneous HEV-3f strains, phylogenetic trees were constructed based on 442-nucleotide ORF1 (position 4270 nt to 4713 nt) and 1305-nucleotide ORF2 (position 6269 nt to 7573 nt) sequences. All sequences of HEV-3 strains from South American countries and Brazil, from both human and animal origin, were included in the analysis. However, it is worth mentioning that complete Brazilian HEV-3 genome sequences were not available and only partial Brazilian ORF2 sequences could be used in the phylogenetic analysis. Although partial ORF1 and ORF2 sequences from Brazilian samples, including those from swine origin [59,60], were available from the GenBank database, the molecular studies used different primer sets that amplified distinct portions of the ORFs, thus hampering their alignment and the comparison with the IAL-HEV_921 strain detected here.

The phylogenetic trees obtained for the partial ORF1 and ORF2 regions supported the genotype found in IAL-HEV_921 strain (Figure 1A,B). The nucleotide sequences of the IAL-HEV_921 strain and the HEV-3f strains from Europe and Asia, which had been identified in humans and swine, including domestic pigs and wild boars, had 88.1–93.2% and 89.6–95.3% of nucleotide identity for ORF1 and ORF2, respectively. It is worth mentioning that subtype 3d in the ORF1and ORF2 trees is missing. Previous lineage designations were kept in accordance with Smith et al. [15], Nicot et al. [56,57], and Sarchese et al. [58], and there was no subtype 3d designation in the phylogenetic analysis schemes.

The genetic analysis of the ORF1 and ORF2 regions revealed that the Brazilian capybara IAL-HEV_921 strain clustered with a wild boar WB03VT2016 strain detected in Italy in 2016 (90.7–93.2% nt), a swine FR-SHEV3f strain detected in France in 2008 (91.4–92.4% nt) and two human strains: the NL_Donor_48 strain detected in 2015 in The Netherlands and the HESQL097 strain detected in 2018 in France (91.3–92.7% nt) [57] (Figure 1A,B). The ORF2 sequence of the Brazilian capybara IAL-HEV_921 strain detected here distantly clustered with HEV-3 sequences previously detected in Brazil. Human HEV-3 Brazilian strain Brazilh4 and swine HEV-3 Brazilian strains sw8HEVBR, wHEVBR-108-02 and LBZ10hev were classified as subtype HEV-3b; while monkey HEV-3 strain CY-LAC7 and swine HEV-3 Brazilian strains swMG314, S5VIT and SW-IS7 subtypes could not be assigned (Figure 1B).

## 4. Discussion

This study documents the genetic characterization of a hepatitis E virus genotype 3 strain detected for the first time in capybaras. HEV-3 is known to have a zoonotic origin and has been reported worldwide, being the predominant genotype in Europe and America [57]. In fact, HEV-3 is the only reported genotype in humans, animals and environments in Brazil [61].

HEV typing and subtyping is crucial in determining the transmission route and chain of infection as well as improving our understanding of their potential health burden. Ideally, full HEV genomes should be used to analyze recombination events and to assign reliable novel genotypes and subtypes [57]. A limitation of the present report was the failure to obtain a full IAL-HEV_921 genome. Direct identification of viral genomes from clinical specimens using NGS remains challenging, mostly due to the scarcity of viral genomic material, small size of the virus genome or their low abundance in a high background of other microbial nucleic acids [62]. The low HEV concentration in the capybara fecal sample may therefore explain the failure of full genome sequencing. Facing this limitation, we used both ORF1 and ORF2 genes for phylogenetic analyses and classification in order to guarantee consistency. It has been reported that phylogenetic analyses conducted with the HEV ORF1 gene correlated well with the results from the phylogenetic analysis based on the complete HEV genome [63]. On the other hand, the HEV ORF2 gene was found to be more suitable for determining HEV genotype and subtype [64].

The topology of the phylogenetic trees obtained from ORF1 and ORF2 sequences sustained the IAL-HEV_921 strain genetic relatedness as well as its subtype assigned as HEV-3f. The IAL-HEV_921 strain was closely related to European swine, wild boar and human strains. The close relationship between HEV strains isolated from different host species points towards interspecies transmission [65]. The poor conclusions between IAL-HEV_921 strain and other Brazilian HEV-3 strains drawn from the phylogenetic analyses reflect the paucity of data available in the national literature regarding HEV-3 sequences, as well as the failure to align with cognate swine HEV-3f partial sequences. However, it is worth mentioning that former studies conducted in Brazil addressing molecular surveillance of swine HEV-3 strains have identified genotypes 3a, 3b, 3c, 3d, 3h, 3i and 3f circulating in the country [60,65,66,67,68,69,70,71].

Subtype HEV-3f is attracting attention in Europe. This subtype has been assigned to acute human HEV cases in France [72], high hospitalization risk in Belgium [73] and has been associated with a hepatitis E outbreak in Italy in 2019 [74]. Our knowledge concerning autochthonous HEV-3f infection in Brazil is very limited. HEV is not commonly investigated in Brazil and there is a gap in the understanding of its molecular epidemiology [61]. So far, subtype 3f has only been detected in swine in Brazil [60,65,66,67,68,69,70,71]. It can be speculated that HEV-3f may have been silently circulating in humans, because HEV infections are commonly asymptomatic or subclinical and there is no continuous laboratory-based surveillance for HEV in the Brazilian population, including genotyping and subtyping [75].

The importance of wild rodents as potential HEV reservoirs or their zoonotic role is still unknown [15,34,35,36,37]. The detection of HEV in capybaras must be treated with caution. Identification of HEV in these mammals may not necessarily be associated with natural infection and may not indicate that these animals play a role in the transmission of HEV as the virus detected in feces could have originated from environmental contamination. It is important to mention that aside from anthropogenic influences, the local water bodies that capybaras inhabit are also contaminated with sewage. Microbial analysis conducted by Silva et al. [76] revealed a full contamination of fecal coliforms with 75% of *E. coli* in the Tietê Ecological Park (PET) water bodies due to sewage dumping, mostly domestic. Silva et al. [76] also highlighted the issue of viral contamination once viruses are disseminated through water and can contaminate the entire ecosystem, compromising the local fauna, and HEV is well known to be transmitted mainly via the fecal–oral route through contaminated drinking water [15]. In addition, although very rare, wild boars (i.e., cateto) have already been sighted on the trails across the park (personal communication). Therefore, capybaras are continuously exposed to fecal-borne viruses and to a potential zoonotic-transmission route as these animals are living in human–animal interface environments. To better elucidate the existence of a sylvatic cycle of HEV associated to capybaras, further studies on infectivity are required [34], as well as investigations into the presence of HEV-3f in different local species, including humans and swine, as well as the environment.

It is important to highlight that the capybaras enrolled in the study were free-living animals and the fecal samples were obtained from natural evacuation. No capybara was captured or sacrificed; therefore, one limitation of the present investigation is the lack of liver and serum samples collected from the animals in order to evaluate HEV replication and verify serological evidence of previous infections by HEV.

Knowledge about the presence and diversity of fecal-borne viruses in capybaras is so far limited and a better understanding of the potential role of HEV infections in these animals is essential to develop preventive measures. The present study reports a molecular surveillance of HEV in Brazilian capybaras and adds further evidence that the fecal-borne viruses could constitute a public health issue. Continuous monitoring of sylvatic animals is essential to prevent and control the emergence or re-emergence of newly discovered viruses as well as viruses with known zoonotic potential, especially in anthropogenic environments. Capybaras are extremely tolerant to environmental changes and exhibit high vagility along waterways, thus having potential to trigger emerging zoonosis [52].

In conclusion, this study alerts the need to improve the monitoring of HEV in wildlife throughout Brazil. Zoonotic studies are hampered by lack of genome sequencing data of HEV circulating in animals, especially in Brazil. The implementation of systematic molecular surveillance of HEV in the One Health concept, including human, wildlife, domestic pigs, wild boars and environment samples is vital to elucidate HEV-3 subtypes circulating in the country as well as to acquire further evidence of the zoonotic potential of HEV infections.

## Figures and Tables

**Figure 1 viruses-15-00335-f001:**
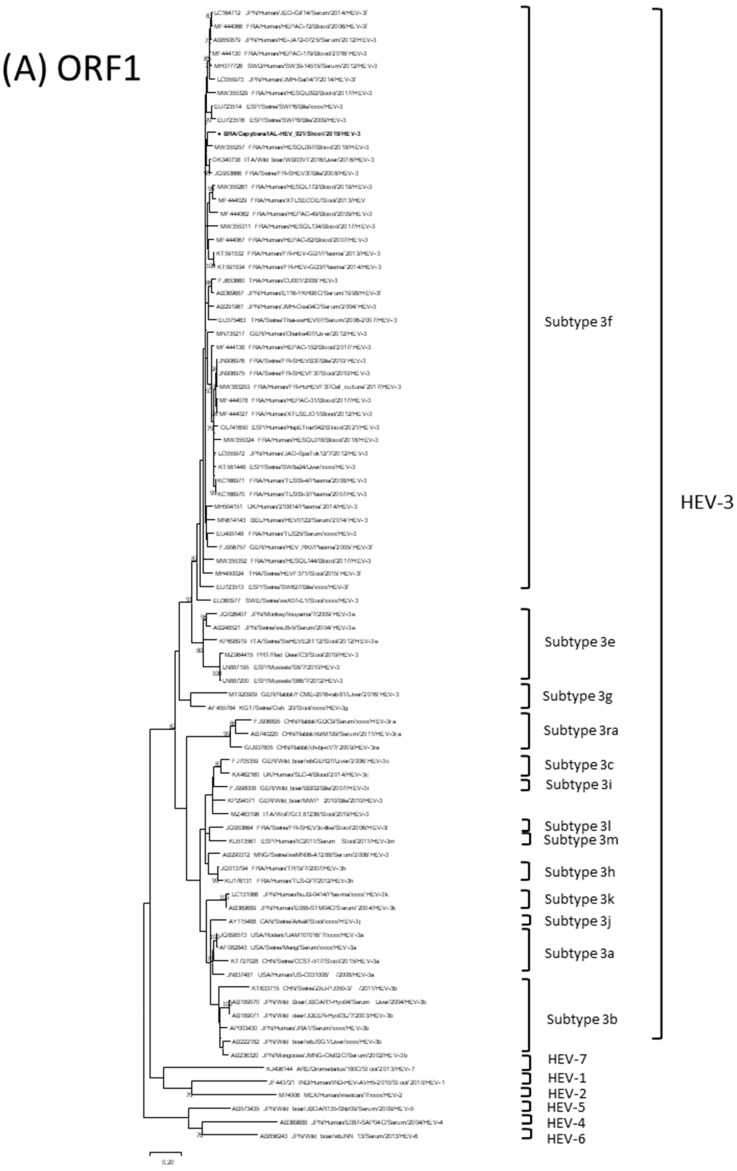
Nucleotide based phylogenetic relatedness of IAL-HEV_921 strains (indicated in bold and by ●) (**A**) ORF1 and (**B**) ORF2 to other selected HEV strains. Maximum likelihood trees of partial nucleotide sequences were generated with MEGA X software versiom 7.0 [55] (megasoftware.net; accessed on 18 October 2022. Reference strains were obtained from the GenBank database. Brazilian HEV-3 strains are indicated by ▲. Genotypes, subtypes, accession number, isolates, countries and year of each strain are indicated. The scale indicates the number of divergent nucleotide residues. Percentages of bootstrap values are shown at the branch node.

## Data Availability

Sequences from the present study were deposited in GenBank under the accession numbers: OP485092 for the ORF1 region and OP485093 for the ORF2 region.

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
