# Peer review of "Detection of Hepatitis E Virus Genotype 3 in Feces of Capybaras (Hydrochoeris hydrochaeris) in Brazil"

_viruses, 2023, doi:10.3390/v15020335_

Round 1

Reviewer 1 Report (Previous Reviewer 1)

Thank you to the Authors for the answers.

After this revision of the manuscript, I suggest the publication in “Viruses” journal.

Author Response

Response to Reviewer’s and Academic Editor’s Comments

Manuscript ID viruses-1998588

Dear Editor,

Thank you for your and the reviewer’s comments, and for the publication consideration. We revised the manuscript, and the list of changes or rebuttal is right after each point rose highlighting on yellow.

Yours sincerely,

Regina Moreira and coworkers

Major comments

  1. Reviewers pointed out that a major limitation of this study was the omission of detection of HEV-C in capybara samples. The authors performed some limited screening for HEV-C in response. Please include these results in the manuscript. Also include this in the limitations of the manuscript that not all capybara samples were screened for HEV-C.

We thank the reviewers for their valuable time in reviewing the manuscript. I believe that we did not clearly answer this question. Here we are improving the answers in order to clarify the point highlighted.

We are so very grateful for the reviewer’s suggestion, and we understand the concerns about HEV-C1 screening. Nevertheless, the fact that we did not assess HEV-C1 presence in capybaras fecal samples does not invalidate the HEV-3f data obtained. The data is valuable for HEV monitoring in sylvatic mammals and completely new and relevant in the HEV-A field.

As explained in the last review, we had not included the screening for HEV-C1 in the original study aims. Our sole objective was screening for HEV-A. We conducted a sampling screening of approximately 15% in order to pacify reviewers worries, and we emphasized that we did not have the financial resources to screen all 337, but assuring that HEV-C1 screening was take into consideration in future studies. We are fully aware that 15% (50 samples) do not represent the sampling reality; after all we had to screen more than 300 samples to get only one positive result for HEV-3f. With all due respect, we reserve the right to make this data public only when we are able to screen all the study samples.

  1. How certain are the authors that the stool samples are actually from a capybara instead of another species? One way the authors could address this is to analyze the NGS data to check the alignment of mammalian reads.

We consider the reviewer might not familiar with capybaras. Capybaras are very distinctive animals and their feces are peculiar, easily recognized. Capybaras can excrete two types of feces, one in the oval form in individualized olive-green pellets and another with a pasty consistency and lighter color. Both stand on easily recognizable mounts. This information was included in the methods item. Please, see lines 117 and 118. Thanks.

  1. Even if from a capybara, the authors must note that simply detecting 3f in stool doesn’t necessarily represent infection. Paslahepevirus balayani has been detected in the guts of rodents without productive infection because of ingestion of contaminated sewage. Therefore, this study has not disproven the dogma that rodents are not susceptible to Paslahepevirus balayani. Include this in the limitations.

We are fully aware of this limitation. This particular discussion was addressed in the originally submitted manuscript, as well as in revision 1. We also discussed (included in both original manuscript and revision 1) that further studies on potential capybaras infection caused by HEV-3f must be conducted, as well as investigations into the presence of HEV-3f in different local species, including humans and swine, as well as the environment. These limitations was highlighted again in red in the revision 2. Please, see lines 307-311.

  1. There are numerous grammatical errors throughout the manuscript. Some are listed below. Please review English language carefully in your revision. This is critical for the manuscript’s acceptance.

English language was revised.

Minor comments

  1. Delete ‘First’ in the title. Ok
  2. Delete ‘an emerging global disease’ in the title. Ok
  3. Abstract Line 14: ‘associated with’, not ‘associated to’. Ok
  4. Abstract Line 23 and line 273: correct to capybara stool samples instead of ‘capybaras stool samples’. Ok
  5. Line 31: amend this sentence. Asia, Middle East and Africa are not countries.
  6. Line 38: the 3 ORFs span the genome, they are not ‘located near the 5’-end’. ok
  7. Line 55, Line 62/63: correct grammar. Ok
  8. Line 73: Europe, not Europa. Ok
  9. Line 123: defined, not ‘define’. Ok
  10. Line 214: replace ‘totalizing’ with totaling. Ok
  11. Line 276 & 333: please review words ‘consentaneity’ and ‘vagility’.

The term “vagility” is correct and means the ability of an organism to move about freely and migrate, very characteristic of capybaras behavior.  

  1. The finding that the strain is subtype 3f is repeated twice in the results. Please edit. Ok
  2. Reference 15 is an update of reference 56. Please only cite reference 15. Ok
  3. The sentence in lines 232-237 should be edited. Ok

Reviewer 2 Report (Previous Reviewer 2)

The paper has been implemented by adding analyses for HEV-C1. It can be accepted as it is

Author Response

Response to Reviewer’s and Academic Editor’s Comments

Manuscript ID viruses-1998588

Dear Editor,

Thank you for your and the reviewer’s comments, and for the publication consideration. We revised the manuscript, and the list of changes or rebuttal is right after each point rose highlighting on yellow.

Yours sincerely,

Regina Moreira and coworkers

Major comments

  1. Reviewers pointed out that a major limitation of this study was the omission of detection of HEV-C in capybara samples. The authors performed some limited screening for HEV-C in response. Please include these results in the manuscript. Also include this in the limitations of the manuscript that not all capybara samples were screened for HEV-C.

We thank the reviewers for their valuable time in reviewing the manuscript. I believe that we did not clearly answer this question. Here we are improving the answers in order to clarify the point highlighted.

We are so very grateful for the reviewer’s suggestion, and we understand the concerns about HEV-C1 screening. Nevertheless, the fact that we did not assess HEV-C1 presence in capybaras fecal samples does not invalidate the HEV-3f data obtained. The data is valuable for HEV monitoring in sylvatic mammals and completely new and relevant in the HEV-A field.

As explained in the last review, we had not included the screening for HEV-C1 in the original study aims. Our sole objective was screening for HEV-A. We conducted a sampling screening of approximately 15% in order to pacify reviewers worries, and we emphasized that we did not have the financial resources to screen all 337, but assuring that HEV-C1 screening was take into consideration in future studies. We are fully aware that 15% (50 samples) do not represent the sampling reality; after all we had to screen more than 300 samples to get only one positive result for HEV-3f. With all due respect, we reserve the right to make this data public only when we are able to screen all the study samples.

  1. How certain are the authors that the stool samples are actually from a capybara instead of another species? One way the authors could address this is to analyze the NGS data to check the alignment of mammalian reads.

We consider the reviewer might not familiar with capybaras.  Capybaras are very distinctive animals and their feces are peculiar, easily recognized. Capybaras can excrete two types of feces, one in the oval form in individualized olive-green pellets and another with a pasty consistency and lighter color. Both stand on easily recognizable mounts. This information was included in the methods item. Please, see lines 117 and 118. Thanks.

  1. Even if from a capybara, the authors must note that simply detecting 3f in stool doesn’t necessarily represent infection. Paslahepevirus balayani has been detected in the guts of rodents without productive infection because of ingestion of contaminated sewage. Therefore, this study has not disproven the dogma that rodents are not susceptible to Paslahepevirus balayani. Include this in the limitations.

We are fully aware of this limitation. This particular discussion was addressed in the originally submitted manuscript, as well as in revision 1. We also discussed (included in both original manuscript and revision 1) that further studies on potential capybaras infection caused by HEV-3f must be conducted, as well as investigations into the presence of HEV-3f in different local species, including humans and swine, as well as the environment. These limitations was highlighted again in red in the revision 2. Please, see lines 307-311.

  1. There are numerous grammatical errors throughout the manuscript. Some are listed below. Please review English language carefully in your revision. This is critical for the manuscript’s acceptance.

English language was revised.

Minor comments

  1. Delete ‘First’ in the title. Ok
  2. Delete ‘an emerging global disease’ in the title. Ok
  3. Abstract Line 14: ‘associated with’, not ‘associated to’. Ok
  4. Abstract Line 23 and line 273: correct to capybara stool samples instead of ‘capybaras stool samples’. Ok
  5. Line 31: amend this sentence. Asia, Middle East and Africa are not countries.
  6. Line 38: the 3 ORFs span the genome, they are not ‘located near the 5’-end’. ok
  7. Line 55, Line 62/63: correct grammar. Ok
  8. Line 73: Europe, not Europa. Ok
  9. Line 123: defined, not ‘define’. Ok
  10. Line 214: replace ‘totalizing’ with totaling. Ok
  11. Line 276 & 333: please review words ‘consentaneity’ and ‘vagility’.

The term “vagility” is correct and means the ability of an organism to move about freely and migrate, very characteristic of capybaras behavior.  

  1. The finding that the strain is subtype 3f is repeated twice in the results. Please edit. Ok
  2. Reference 15 is an update of reference 56. Please only cite reference 15. Ok
  3. The sentence in lines 232-237 should be edited. Ok

Reviewer 3 Report (New Reviewer)

The manuscript has been improved. I recommend accepting the manuscript.

Author Response

Response to Reviewer’s and Academic Editor’s Comments

Manuscript ID viruses-1998588

Dear Editor,

Thank you for your and the reviewer’s comments, and for the publication consideration. We revised the manuscript, and the list of changes or rebuttal is right after each point rose highlighting on yellow.

Yours sincerely,

Regina Moreira and coworkers

Major comments

  1. Reviewers pointed out that a major limitation of this study was the omission of detection of HEV-C in capybara samples. The authors performed some limited screening for HEV-C in response. Please include these results in the manuscript. Also include this in the limitations of the manuscript that not all capybara samples were screened for HEV-C.

We thank the reviewers for their valuable time in reviewing the manuscript. I believe that we did not clearly answer this question. Here we are improving the answers in order to clarify the point highlighted.

We are so very grateful for the reviewer’s suggestion, and we understand the concerns about HEV-C1 screening. Nevertheless, the fact that we did not assess HEV-C1 presence in capybaras fecal samples does not invalidate the HEV-3f data obtained. The data is valuable for HEV monitoring in sylvatic mammals and completely new and relevant in the HEV-A field.

As explained in the last review, we had not included the screening for HEV-C1 in the original study aims. Our sole objective was screening for HEV-A. We conducted a sampling screening of approximately 15% in order to pacify reviewers worries, and we emphasized that we did not have the financial resources to screen all 337, but assuring that HEV-C1 screening was take into consideration in future studies. We are fully aware that 15% (50 samples) do not represent the sampling reality; after all we had to screen more than 300 samples to get only one positive result for HEV-3f. With all due respect, we reserve the right to make this data public only when we are able to screen all the study samples.

  1. How certain are the authors that the stool samples are actually from a capybara instead of another species? One way the authors could address this is to analyze the NGS data to check the alignment of mammalian reads.

We consider the reviewer might not familiar with capybaras.  Capybaras are very distinctive animals and their feces are peculiar, easily recognized. Capybaras can excrete two types of feces, one in the oval form in individualized olive-green pellets and another with a pasty consistency and lighter color. Both stand on easily recognizable mounts. This information was included in the methods item. Please, see lines 117 and 118. Thanks.

  1. Even if from a capybara, the authors must note that simply detecting 3f in stool doesn’t necessarily represent infection. Paslahepevirus balayani has been detected in the guts of rodents without productive infection because of ingestion of contaminated sewage. Therefore, this study has not disproven the dogma that rodents are not susceptible to Paslahepevirus balayani. Include this in the limitations.

We are fully aware of this limitation. This particular discussion was addressed in the originally submitted manuscript, as well as in revision 1. We also discussed (included in both original manuscript and revision 1) that further studies on potential capybaras infection caused by HEV-3f must be conducted, as well as investigations into the presence of HEV-3f in different local species, including humans and swine, as well as the environment. These limitations was highlighted again in red in the revision 2. Please, see lines 307-311.

  1. There are numerous grammatical errors throughout the manuscript. Some are listed below. Please review English language carefully in your revision. This is critical for the manuscript’s acceptance.

English language was revised.

Minor comments

  1. Delete ‘First’ in the title. Ok
  2. Delete ‘an emerging global disease’ in the title. Ok
  3. Abstract Line 14: ‘associated with’, not ‘associated to’. Ok
  4. Abstract Line 23 and line 273: correct to capybara stool samples instead of ‘capybaras stool samples’. Ok
  5. Line 31: amend this sentence. Asia, Middle East and Africa are not countries.
  6. Line 38: the 3 ORFs span the genome, they are not ‘located near the 5’-end’. ok
  7. Line 55, Line 62/63: correct grammar. Ok
  8. Line 73: Europe, not Europa. Ok
  9. Line 123: defined, not ‘define’. Ok
  10. Line 214: replace ‘totalizing’ with totaling. Ok
  11. Line 276 & 333: please review words ‘consentaneity’ and ‘vagility’.

The term “vagility” is correct and means the ability of an organism to move about freely and migrate, very characteristic of capybaras behavior.  

  1. The finding that the strain is subtype 3f is repeated twice in the results. Please edit. Ok
  2. Reference 15 is an update of reference 56. Please only cite reference 15. Ok
  3. The sentence in lines 232-237 should be edited. Ok

This manuscript is a resubmission of an earlier submission. The following is a list of the peer review reports and author responses from that submission.

Round 1

Reviewer 1 Report

Cunha et al., in the article “First detection of hepatitis E virus genotype 3 in feces of capybaras (Hydrochoeris hydrochaeris) in Brazil: an emerging global disease” described the results of molecular survey aimed to investigate the presence of HEV RNA in free living-capybaras in urban parks in São Paulo state, Brazil. A stool sample resulted positive, and the strain detected was assigned to the subtype HEV 3f. The manuscript provides new data about HEV circulation in a geographical setting for which to date information are still limited.

In the introduction, and all over the manuscript, the Authors must use the most recent classification for the family Hepeviridae (Purdy et al., 2022).

Line 31: Please replace HEV with Hepatitis E

Line 33: Please note that HEV could be also a quasi-enveloped virus

Lines 47-50: Please rephase the sentence.

Line 57: The references described the identification of HEV3 in rat but also of strains classified within the genus Rocahepevirus, species Rocahepevirus ratti, genotype C1. The Authors should explain clearer this point. In addition, they should introduce this genotype, considering that 1) capybaras are rodents and HEV-C1 are the most frequent genotype detected in Rodentia 2) HEV-C1 are pathogenic for humans (Sridhar et al., 2018, Andonov et al., 2019, Reuter et al., 2020, Sridhar et al., 2021, Rivero-Juarez et al., 2022).

A molecular investigation aimed to detected HEV classified within the species Paslahepevirus balayani was performed. Did the Author try to assess the samples also for Rocahepevirus ratti, genotype C1?

Could the Authors discuss the implication to detect Paslahepevirus balayani in this rodent species?

Reviewer 2 Report

The paper deals with the detection of the zoonotic HEV-3 in capybaras (Hydrochoeris hydrochaeris). The novelty of the paper is the first detection of HEV in this animal host. However, a major limit of the study is that HEV-C was not investigated but only HEV-A . In the last few years, the zoonotic potential of HEV-C is largely discussed due to detection of the genotype HEV-C1 is humans. Due to this, besides HEV-A also HEV-C should be investigated by the authors and the paper should also aim to evaluate the circulation of HEV-C. To this purpose the same RNA should be further tested for HEV-C.

Besides this the paper is well written and results well supported, but can not be accepted for publication before HEV-C will be investigated.

The introduction should be revised by adding information on HEV-C in rats. Recently, HEV rat strains have been described in human cases, suggesting that Orthohepevirus species C could be also zoonotic specie hosted by rats. There are several papers available on this topic.

Furthermore, for the same reason the authors should test RNA from rats using a broad range RT-PCR not only the Real-time that is specifically designed for HEV-1-4 genotypes. The protocol they used for sequencing ORF1 by Johne et al., 2010 can also be used for HEV-C1. I suggest to analyse all samples with both protocols Real-time (HEV1-4) and RT-PCR (Johne et al., 2010) to screen samples for both HEVs, HEV-A and HEV-C

 Line 38-41: please update the new classification; see the recently published paper https://doi.org/10.1099/jgv.0.001778

line 56: please add more references and countries such as UK Murphy et al., 2019; IT De Sabato et al., 2019, Zanet e tal., 2019

line 85: what does “obtained in natura” mean? Were the feces collected on the floor or in cages? How do the authors can confirm that one feces corresponds to one animal?

line 91-93: could the authors add details about the Real-time, such as kit used. The protocol by Jothikumar et al is widely used, but it is designed only for HEV-A. The authors should better specify their choice in the text

Line 94: some controls for animals are available but there are not controls for capybaras, please modify your statement accordingly.

Line 162-163: the authors should revise the subtypes assignment. In the paper, by Smith et al., 2020, that is the reference for subtypes assignment subtypes from a to letter m are named and some additional subtypes are proposed but not named yet for the lack of genome sequence,

In the phylogenetic trees proposed by the authors there could be some mistakes in the list of subtypes. First, the subtype proposed by Smith et al (ICTV reference paper) is up to the letter m and the authors proposed also q and r.  Second, some references are not correct, for example the JQ953664 is the reference of 3l, please check them.

RESULTS are properly written and I have no comments, but I’d suggest considering the HEV-C detection to complete findings of the study.

Discussion, since the HEV-3 was only detected in feces, it is unknown if the virus can replicate in the capybaras. In fact, the presence of HEV-RNA could be only due to consumption of contaminated feces by capybaras (Hydrochoeris hydrochaeris). To evaluate replication at least liver should have been assayed. The authors should clearly state it in the discussion

Reviewer 3 Report

Introduction section: Authors should correct their citations. Most references are not related to the context. For instances:

Pages 38-42: HEV taxonomy: Wang et al., 2018 is not the correct citation since the classification of HEV into genus Orthohepevirus A was held in 2012 (Virus Taxonomy 9th Report of the ICTV; Elsevier Academic Press: London, UK, 2012; pp. 1021–1028). Also, subtypes of the species Orthovirus A were reported by Smith et al., 2020.

Pages 50-53: Passos-Castilho et al., 2017 and Heldt et al., 2016 are not the first research group to report the HEV prevalence in swine. Nevertheless, dos Santos DR et al., 2010 and other research groups investigated HEV infection and seroprevalence in Brazilian swine, e.g., Gardinali et al. 2011, de Oliveira Filho et al., 2017, Amorin et al., 2018, among others.

Pages 230-231: Moraes et al., 2021 is a Review. Authors frequently made "Appud" references despite the original articles.

It is strongly recommended that authors verify the References and citations. Many references are unnecessary, and others are missing.

Results:

Pages 187-190: Why did authors not include HEV-3f sequences from Brazilian swine to investigate the genetic relationships among contemporaneous HEV-3f strains? At least two papers have reported HEV-3f in Brazilian swine herds.

Pages 247-259: Indeed, very few Brazilian human HEV sequences are available at GenBank. However, the genetic analysis did not include sequences of the first report of an acute HEV infection in Brazil. Besides, Brazilian swine HEV 3f sequences were not included in the ORF1 and ORF2 phylogenetic trees. It is reasonable to repeat the phylogenetic analysis, including all Brazilian swine HEV-3f and Brazilian human HEV-3 available.

Pages 251-253: I am afraid I must disagree with the assumption that "The poor conclusions between IAL-HEV_921 strain and other Brazilian HEV-3 strains drawn from the phylogenetic analyses reflect the paucity of data available in the national literature regarding HEV-3 sequences". As mentioned above, the authors did not consider other Brazilian HEV-3 sequences. Moreover, the authors did not bring other data from the Brazilian literature for the Discussion in the present manuscript.

The main concern is that the phylogenetic analysis possibly underestimated the swine HEV-3 sequences available from the five Brazilian geographic regions.